Molecular Biology and Physiology

# Comparative Transcriptomic and Functional Assessments of Linezolid-Responsive Small RNA Genes in *Staphylococcus aureus*

Wei Gao,[a] Romain Guérillot,[a] Ya Hsun Lin,[a] Jai Tree,[b] Marie Beaume,[c]* Patrice François,[c] Ian R. Monk,[a] Torsten Seemann,[a] Jacques Schrenzel,[c] Benjamin P. Howden,[a] Timothy P. Stinear[a]

[a]Department of Microbiology and Immunology, Doherty Institute for Infection and Immunity, University of Melbourne, Melbourne, Australia
[b]School of Biotechnology and Biological Sciences, University of New South Wales, Sydney, Australia
[c]Genomic Research Laboratory, Division of Infectious Diseases, Geneva University Hospitals, Geneva, Switzerland

Benjamin P. Howden and Timothy P. Stinear are joint senior authors.

**ABSTRACT** *Staphylococcus aureus* contains a repertoire of at least 50 and possibly 500 small RNAs (sRNAs). The functions of most sRNAs are not understood, although some are known to respond to environmental changes, including the presence of antibiotics. Here, in an effort to better understand the roles of sRNAs in the context of antibiotic exposure, we took a clinical methicillin-resistant *S. aureus* (MRSA) isolate and separately deleted eight sRNAs that were significantly upregulated in response to the last-line antibiotic linezolid as revealed by transcriptome sequencing (RNA-seq) comparisons. We also deleted an additional 10 sRNAs that were either highly expressed or previously found to respond to antibiotic exposure. There were no significant changes for any of the 18 mutants in a variety of phenotypic screens, including MIC screens, growth competition assays in the presence of linezolid, biofilm formation, and resistance to whole-blood killing. These data suggest sRNA functional redundancy, because despite their high expression levels upon antibiotic exposure, individual sRNA genes do not affect readily observable bacterial phenotypes. The sRNA transcriptional changes we measured during antibiotic exposure might also reflect sRNA "indifference," that is, a general stress response not specifically related to sRNA function. These data underscore the need for sensitive assays and new approaches to try and decipher the functions of sRNA genes in *S. aureus*.

**IMPORTANCE** Bacterial small RNAs (sRNAs) are RNA molecules that can have important regulatory roles across gene expression networks. There is a growing understanding of the scope and potential breadth of impact of sRNAs on global gene expression patterns in *Staphylococcus aureus*, a major human pathogen. Here, transcriptome comparisons were used to examine the roles of sRNA genes with a potential role in the response of *S. aureus* to antibiotic exposure. Although no measurable impact on key bacterial phenotypes was observed after deleting each of 18 sRNAs identified by these comparisons, this research is significant because it underscores the subtle modes of action of these sometimes abundant molecules within the bacterium.

**KEYWORDS** RNA sequencing, *Staphylococcus aureus*, mutagenesis, postantibiotic effects, sRNA, transcriptome

*S*taphylococcus aureus* is an opportunistic human pathogen causing widespread hospital- and community-acquired infections (1). Expression of virulence factors and acquired antibiotic resistance are key factors in the pathogenesis of *S. aureus*. Small RNAs (sRNAs) are short, usually noncoding regulatory RNA molecules, and they provide

This article followed an open peer review process. The review history can be read here.

Address correspondence to Timothy P. Stinear, tstinear@unimelb.edu.au.

* Present address: Marie Beaume, Deinove, Cap Sigma/ZAC Euromédecine II, Grabels, France.

additional levels of control to regulatory networks, contributing to the flexibility and dynamics of target gene expression, often by sensing environmental changes (2, 3). The *cis*-acting riboswitches and antisense sRNAs are encoded on the cDNA strand of protein-coding genes, whereas most *trans*-acting sRNAs are located within conserved intergenic regions (4, 5). Most of the *trans*-acting sRNAs in *S. aureus* regulate their target genes by binding to the target mRNAs and indirectly affecting expression of gene products. Small RNAs can target both the 5′ untranslated regions (5′UTR) and 3′UTR of mRNAs by competing with the ribosome binding site (RBS) and affecting mRNA stability, respectively (5).

A number of studies have helped catalogue a growing repertoire of sRNA genes in *S. aureus*, accelerated by the development of computational prediction tools and DNA sequencing technologies (6–15). Staphylococcal sRNAs have been aggregated into a database that currently includes 607 unique sRNAs for reference ST239 *S. aureus* genome JKD6008 (16) (http://srd.genouest.org/browse/JKD6008). Application of stricter definitions for bona fide sRNAs based on transcript assessments and defined as not *cis*-acting and not antisense molecules suggests that the actual number may be closer to 50 sRNAs. While the discovery and description of new *S. aureus* sRNAs have blossomed, research assigning a function(s) to these molecules has lagged behind. Besides the *cis*-acting riboswitches, one of the most known and well-studied sRNAs in *S. aureus* is RNAIII, a critical virulence factor regulator (17). RNAIII is involved in potentiating at least 12 different mRNAs, decreasing virulence factor expression, such as for gamma hemolysin in early stages of cell invasion and increasing in later stages of infection (18). Another sRNA, *Psm-mec* (Teg4), contributes to activation of biofilm formation, diminishing bacterial virulence, and shapes the transcription profile of *S. aureus* during chronic infection and colonization (19, 20). Interestingly, this sRNA is carried on the staphylococcal cassette chromosome (SCC*mec*), the locus required for methicillin resistance. The sRNA SprX contributes to glycopeptide resistance in *S. aureus* by downregulation of stage V sporulation protein G, SpoVG (21). SprD, a small pathogenicity island RNA, promotes infections in a murine model. It increases the translation of Sbi immune evasion molecules, and consequently weakens the innate and adaptive immune responses (22, 23). In addition to these examples, only a handful of sRNAs have been assessed for functions, including identifying their targets (6, 7, 24–30).

In previous research, we have shown how the global expression profile of sRNAs change in response to antibiotic exposure (31). Here, we explored in more detail the role of regulated intergenic *trans*-encoded sRNAs that are differentially expressed after linezolid exposure. We profiled and compared the transcriptomes with and without linezolid exposure. We selected 18 sRNAs and constructed unmarked sRNA deletion mutants in a clinical *S. aureus* strain, eight of which responded to linezolid exposure as revealed by transcriptome sequencing (RNA-seq) (cutoff of 1.5-fold, $P < 0.05$) and 10 sRNAs that were highly expressed or found previously to respond to antibiotic exposure.

## RESULTS

**Linezolid exposure has a global impact on gene expression.** We first examined the global gene expression response of methicillin-resistant *S. aureus* (MRSA) to the last-line antibiotic linezolid (30 min, 0.5× MIC). RNA-seq analysis revealed 30 sRNAs that were upregulated and 36 sRNAs that were downregulated (see Table S2 in the supplemental material); in addition, there were 602 protein-coding genes (coding sequences [CDS]) upregulated and 499 downregulated (Fig. 1A to C). We also used Northern blotting to confirm the presence of two differentially regulated sRNAs (Fig. 1D and E). Most of the upregulated CDS were related to anabolic and phosphate transport pathways. Linezolid binds the peptidyl transferase center of ribosomes and inhibits protein synthesis (32). Consistent with this mechanism of action, 53 genes encoding ribosomal proteins were upregulated after antibiotic exposure, indicating that *S. aureus* compensates for translation inhibition by increasing ribosomal protein synthesis (Fig. 1B; see Fig. S1 and Table S2 in the supplemental material). Among the significantly

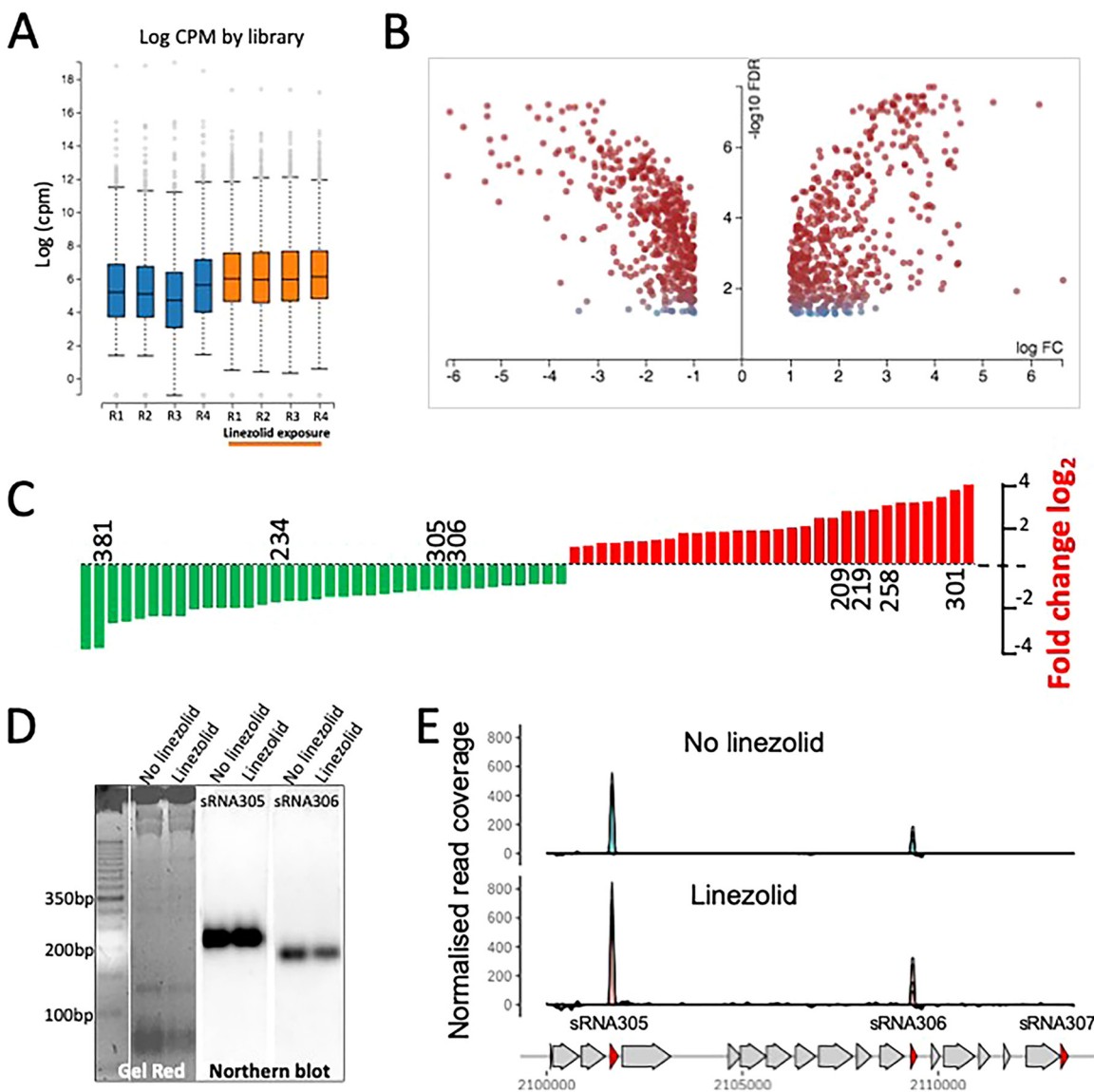

**FIG 1** RNA-seq analysis of *S. aureus* transcriptome with linezolid exposure (0.5× MIC, 30 min). (A) Box-and-whisker plots of normalized read counts (counts per million [cpm]) for each of the eight libraries. (B) Volcano plot showing differentially expressed genes (sRNA and CDS). Red dots represent significant expression changes (twofold change, adjusted $P < 0.05$). FC, fold change. (C) Summary of differentially expressed sRNA with the expression difference for eight sRNAs selected for deletion by the indicated allelic exchange. (D) Confirmatory Northern blot analysis, showing detection of two putative sRNAs (sRNA305 [213 bp] and sRNA306 [165 bp]) identified by RNA-seq. (E) RNA-seq normalized coverage plots for sRNA305 and sRNA306, showing chromosomal locations in *S. aureus* JKD6009.

upregulated sRNAs, 15 were located within *S. aureus* conserved chromosome regions based on the staphylococcal regulatory RNA database (SRD) (16), including *sprX*, which regulates *spoVG* and is involved in glycopeptide resistance (21), suggesting that *S. aureus* exposure to linezolid might have consequences for glycopeptide susceptibility. Downregulated sRNAs included sRNA381 (*rsaOT*), which has previously been shown to increase expression under oxidative stress (26), an indication that the *S. aureus* response to linezolid is distinct from the response that occurs during oxidative stress (Fig. S1).

**Selection of 18 sRNA genes for deletion.** Informed by the transcriptome analysis above and preceding research, we next established a panel of 18 *S. aureus* sRNA unmarked deletion mutants using our previously described ST239 MRSA strain JKD6009. We then compared these mutants to wild-type JKD6009 in a variety of phenotypic assays. We selected sRNA genes that were either (i) differentially expressed

**TABLE 1** sRNAs and mutants examined in this study

| sRNA name | sRNA database name | Justification for inclusion[a] | sRNA knockout mutant | sRNA reference(s) |
|---|---|---|---|---|
| sRNA389 | srn_4830 | High expression | BPH1338 | 14, 31 |
| sRNA258 | srn_9320 | Lz responsive | BPH1349 | 31 |
| sRNA293 | srn_9360 | High expression | BPH1351 | 31 |
| sRNA363 | srn_4470 | High expression | BPH1354 | 7, 15, 31 |
| sRNA406 | srn_5070 | Regulates *icaR* | BPH1356 | 15 |
| sRNA381 | srn_4670 | Lz responsive | BPH1359 | 15, 26, 31 |
| sRNA131 | srn_1490 | High expression | BPH1541 | 14, 15, 26 |
| sRNA219 | srn_2660 | Lz responsive | BPH1350 | 31 |
| sRNA234 | srn_2950 | Lz responsive | BPH1547 | 15 |
| sRNA254 | srn_3210 | High expression | BPH1550 | 31 |
| sRNA259 | srn_3270 | Abx responsive[b] | BPH1553 | 31 |
| sRNA264 | srn_3320 | High expression | BPH1557 | 31 |
| sRNA301 | srn_3790 | Lz responsive | BPH1558 | 31 |
| sRNA352 | srn_4340 | High expression | BPH1560 | 15 |
| sRNA209 | srn_2530 | Lz responsive | BPH1566 | 31 |
| sRNA400 | srn_5010 | High expression | BPH1571 | 15 |
| sRNA305 | srn_1578 | Lz responsive | BPH1578 | 31; this study[c] |
| sRNA306 | srn_1580 | Lz responsive | BPH1580 | 31; this study[c] |

[a]Lz, linezolid; Abx, antibiotic.
[b]Note that in a previous study, this sRNA was shown responsive to fifth-generation cephalosporin exposure (31).
[c]Confirmed by Northern blotting in this study.

with linezolid exposure, (ii) highly expressed in general, or (iii) previously reported to be linked to an antibiotic response (Table 1 and Fig. S1).

**The selected sRNAs do not affect biofilm formation.** Biofilm formation is an important *S. aureus* virulence factor. We noticed that RsaX25 (sRNA406), the 3′UTR of *icaR*, as assessed by RNA-seq was moderately upregulated (1.67-fold) with exposure to linezolid. It has been reported that the 3′UTR of *icaR* can stabilize the mRNA of *icaR* and consequently generate more IcaR, the repressor of *ica* operon (33). Therefore, we decided to test the biofilm formation potential of all sRNA mutants. With three different growth conditions, there was no significant difference in biofilm formation for any mutant compared to wild-type JKD6009 (Fig. 2).

**No impact of sRNA loss in a whole-blood killing assay.** The *S. aureus* sRNA SprD is associated with virulence by regulating the expression of an immune evasion molecule (23). Here, we speculated that sRNAs regulated by linezolid exposure might relate to persistence in blood. Using a whole-blood killing assay, we observed that around 50% of the initial bacterial population survived, but there was no significant

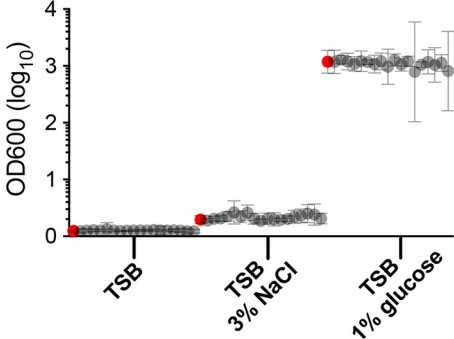

**FIG 2** Biofilm formation of the 18 sRNA deletion mutants in different media compared to wild type. Biofilm formation of the sRNA deletion mutants and wild type in three different media, TSB only, TSB with 3% NaCl, and TSB with 1% glucose, is shown. Depicted are the mean and 95% confidence interval (CI) (error bars) based on three biological replicates for wild-type *S. aureus* JKD6009 (red circles) and all mutants in all three conditions.

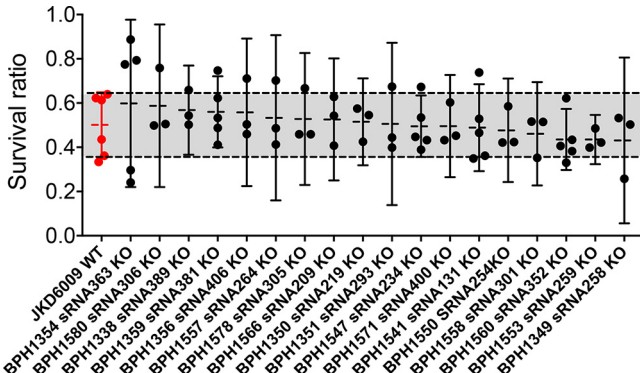

**FIG 3** Whole-blood killing assay showing that survival ratios were not different between the wild-type *S. aureus* JKD6009 and 18 sRNA knockout mutants. Each isolate was tested independently at least three times with the wild-type (WT) strain JKD6009 (red) as a reference. Depicted are the mean survival ratio and 95% confidence intervals (CI) for each knockout (KO) mutant. The gray-shaded area shows the 95% CI survival response for the wild type.

difference observed between any of the sRNA knockout mutants and wild-type JKD6009 (Fig. 3).

**No impact of sRNA loss on susceptibility to antibiotics.** We profiled the antibiograms of all our sRNA mutants to assess their impact on antibiotic resistance. The Vitek results showed that antibiotic susceptibilities were not changed for any of the sRNA knockout mutants (Table S3). There was concordance between Vitek macro E-tests for vancomycin and linezolid susceptibilities. The inducible clindamycin resistance was also verified by D-test, and no difference was found between mutants and wild type. We also conducted linezolid time-kill assays, and these also showed no significant difference in CFU reduction between the wild-type JKD6009 strain and the sRNA knockout mutants after exposure to 4 mg/liter linezolid for 24 h (Fig. 4A).

We then selected sRNAs with significant changes in gene expression with linezolid exposure (sRNA234 and sRNA258) and tested the deletion mutants in competitive growth assays against the wild type. Encouragingly, we observed that the sRNA knockout mutants were significantly outcompeted when grown in the presence of linezolid. However, when the sRNA deletion in each mutant was repaired, no competitive difference was detected when each repaired strain was then competed against its sRNA deletion mutant. The differences observed might therefore be due to secondary mutations unintentionally introduced during the allelic exchange to create the sRNA deletions (Fig. 4B and C). To test this hypothesis, we sequenced the genomes of both mutants and for BPH1349 (sRNA258 knockout) found a missense mutation in JKD6008_00748 (Glu225Gly). JKD6008_00748 encodes SstA, an ABC iron transporter permease (34). BPH1547 (sRNA258 knockout) had a frameshift mutation in *blaR1*. Both of these secondary mutations could conceivably explain the competitive growth defects of these mutants.

**Impact of sRNA loss on growth of *S. aureus* in the presence of linezolid.** We established growth curves in rich media for all mutants with increasing concentrations of linezolid. We then derived doubling times to try and identify sRNA deletion mutants that had doubling times significantly different from the doubling time of wild-type JKD6009 during growth in the presence of 2 ng/ml linezolid. Under these conditions, none of the 18 mutants had growth rate defects compared to wild-type *S. aureus* (Fig. 5).

## DISCUSSION

Hundreds of potential *S. aureus* sRNAs have been identified across different conditions in several strains; however, most sRNAs remain of unknown function (35). Here, we report on sRNAs in *S. aureus* strain JKD6009 that alter their expression in response to certain antibiotics yet do not directly impact antibiotic susceptibility. It has been

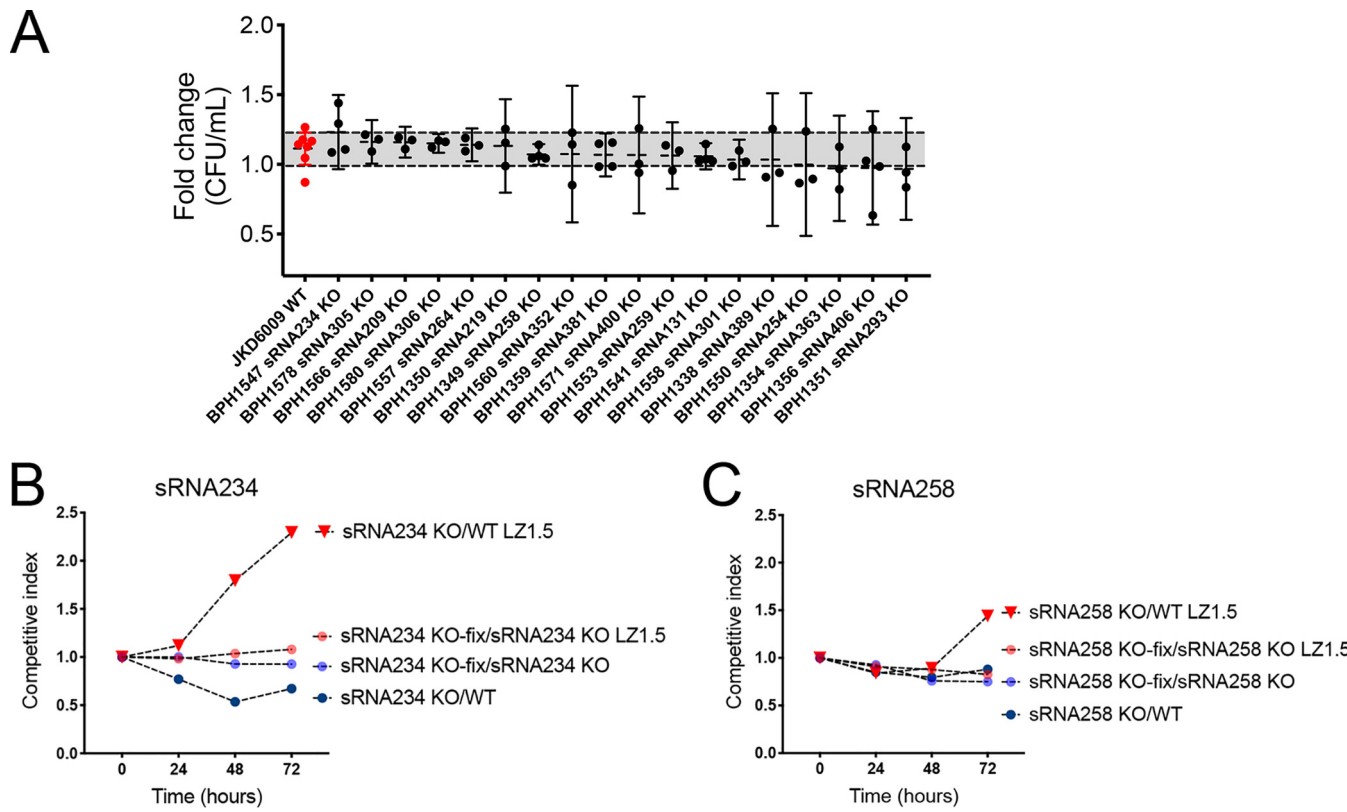

**FIG 4** Assessment of *S. aureus* sRNA roles in response to linezolid. (A) Time-kill assay in 4-mg/liter linezolid BHI for 24 h for 18 *S. aureus* sRNA deletion mutants, showing no significant difference between mutants and wild-type JKD6009. All assays were repeated thrice with independent cultures. Error bars depict 95% CI. Broken lines and gray-shaded area show 95% CI for strain JKD6009. The *y* axis shows fold change in CFU survival compared to wild type. (B and C) Competition assays with two sRNA mutants, complemented (repaired ["KO-fix"]) mutants and wild-type JKD6009. Plots show that the reduced competitiveness observed for the two mutants in the presence of wild-type *S. aureus* and 1.5 mg/liter linezolid was not attributable to the loss of each sRNA gene, as the mutants did not outcompete the repaired strains in the presence of linezolid.

reported that sRNAs are more enriched in the conserved intergenic regions than in the nonconserved regions. The sRNAs selected in our study were all from conserved *S. aureus* genomic regions or genomic regions conserved in other staphylococci. They were either highly expressed or significantly differentially regulated during linezolid

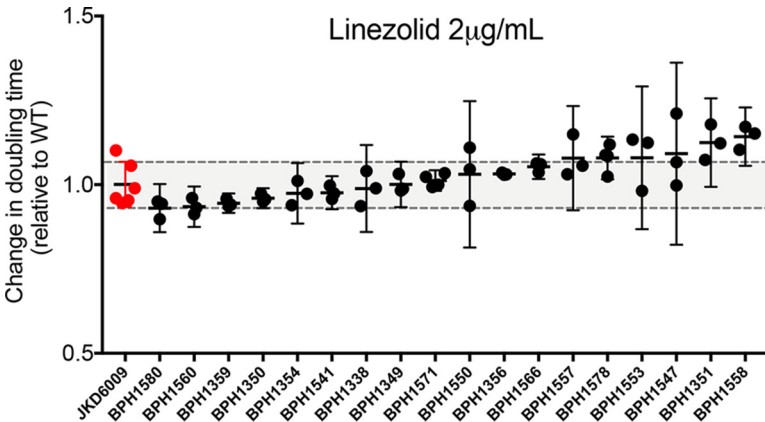

**FIG 5** Comparison of doubling times for the *S. aureus* JKD6009 sRNA deletion mutants compared to the wild type. Growth curves were conducted in TSB with 2.0 μg/ml concentrations of linezolid. Data points indicate results from at least three independent experiments. None of the doubling times were significantly different from values for the wild type. Error bars depict 95% CI. The null hypothesis (no difference between means) was rejected for $P < 0.05$ (Mann-Whitney U-test, unpaired, two tailed).

exposure. We tried to understand the functions of these conserved sRNAs by generating corresponding sRNA deletion mutants. However, the results showed no significant phenotype changes in any of the sRNA mutants constructed. It may indicate that the sRNAs act in more-complex networks that can compensate for the loss of an individual sRNA. The functional redundancy of some bacterial sRNAs has also been reported due to sharing of the same sRNA target (36).

The functions of sRNAs might depend on the specific bacterial genetic background. It has been shown that the activation of *icaR*, the repressor of *icaADBC* operon, is related to biofilm reduction (37). The 3′UTR of *icaR*, sRNA406, inhibits the transcription of *icaR*, and therefore positively regulates the *icaADBC* operon (38). Our inability to detect a change in the biofilm phenotype (with or without supplements) might reflect poor sensitivity of the phenotype assay or might reflect the possibility that this phenotype varies in *S. aureus* strains. It is suggested that sRNA molecules exert their greatest impact at the level of gene transcription; thus, gene expression changes might not always lead to translational changes and observable phenotypes. This phenomenon might indicate that a large proportion of the sRNA repertoire provides transcriptional fine-tuning functions, with phenotypic impacts in some very specific and particular conditions hardly measurable *in vitro*.

We have shown here that sRNAs regulated in response to antibiotic exposure do not directly impact bacterial cell growth and antibiotic resistance *in vitro*. Without knowing the target mRNA binding sites of sRNAs, it is very difficult to predict impact on a specific phenotype or physiological state. Therefore, systematically characterizing the sRNAs and mRNA interactions in *S. aureus* becomes critical to understand the impact of sRNAs to the bacterial pathogenesis. Here, with our panel of assays, we found no significant phenotype changes in the absence of specific sRNAs. The observations demonstrated that some conserved sRNAs across *S. aureus* lineages have no significant contributions to bacterial fitness under certain conditions, including antibiotic exposure. It reflects the sophistication and flexibility of regulons that use sRNAs, wherein sRNAs might be redundant under certain conditions. Furthermore, sRNA-level regulation might be compensated for or overridden by other intersecting regulatory networks. The relatively limited scope of the phenotypic assessments undertaken in this project probably restricts the discovery of the true impact/role of these sRNAs. In future work, transcriptional-level assays and global assessment of sRNA-mRNA hybrids ought to be conducted to characterize sRNAs and their potential interactions in order to identify the molecular targets of specific sRNAs and decipher their functions.

## MATERIALS AND METHODS

**Strains and growth conditions.** The clinical methicillin-resistant *S. aureus* (MRSA) strain JKD6009 (ST239, isolated from a case of bacteremia) and the clean sRNA deletion mutants constructed in this study are listed in Table 1. Brain heart infusion (BHI) (Becton Dickinson) broth or tryptone soy broth (TSB) (Oxoid) were used to culture bacteria. The cultures were incubated aerobically at 37°C with agitation.

**RNA sequencing and data analysis.** The overnight broth cultures were diluted 10 times into prewarmed fresh culture and incubated for around 45 min. Total RNA was extracted from *S. aureus* strain JKD6009 once the culture grown in Mueller-Hinton (MH) broth had reached an optical density at 600 nm ($OD_{600}$) of 0.5 and had been exposed to 0.5× MIC (1 mg/liter) linezolid for 30 min. TRIzol (Invitrogen) and ZymoRNA clean and concentrator columns (ZymoResearch) were used to extract and purify total bacterial RNA following the recommended protocols. rRNA was depleted using Illumina RiboZero rRNA removal kit. The remaining RNA was reverse transcribed to cDNA using Superscript II (Invitrogen). The cDNA library was prepared and sequenced using an Illumina platform as described previously (15). Total RNA from untreated samples was also extracted and sequenced. RNA was extracted from four independent, biological replicate cultures for each condition. Sequence read abundance between the linezolid-exposed and unexposed conditions were quantified using *Kallisto*, a *kmer*-based pseudoalignment tool, against our *S. aureus* JKD6008 reference genome and with differential expression analysis performed using *Voom/Limma* from the *BioConductor R* package, visualized using *Degust* (http://degust.erc.monash.edu/). Differential expression analysis was performed on the RNA sequencing reads. Genes and sRNAs were considered significantly differentially regulated if their transcription level changed more than 1.5-fold and they had a false-discovery rate (FDR) of <0.05 in the linezolid-exposed condition compared to the nonexposed condition.

**Quantitative reverse transcription-PCR (RT-PCR) and Northern blotting.** *S. aureus* cDNA prepared as described above was also used to perform quantitative PCR (qPCR) using the Luna universal qPCR master mix (catalog no. M3003S; New England BioLabs [NEB]). The *gyrB* gene was used as an internal

control to perform a $\Delta\Delta C_T$ calculation. Northern blotting was performed using a nonradioactive method as described previously (39), with probes prepared by *in vitro* transcription (see Table S4 in the supplemental material).

**Selection of target sRNAs.** Eighteen annotated intergenic sRNAs in *S. aureus* were selected in this study. Sixteen selected sRNAs were detected previously by transcriptome sequencing (RNA-seq) (staphylococcal regulatory RNA database [SRD] validated list, http://srd.genouest.org/), and a subset of sRNAs was confirmed in this study by Northern blotting. The sRNAs were located in conserved chromosome regions and are therefore likely involved in core bacterial functions. All 18 selected sRNAs had substantial transcription levels, as evaluated by sequence read counts of >30 reads by growth in either MH broth or linezolid-supplemented MH broth. Eight of the selected sRNAs have been discovered and annotated in other *S. aureus* strains, with the remainder so far only annotated in the ST239 MRSA (40).

**Genetic manipulation.** Each sRNA region was analyzed carefully, with potential promoter and terminator sequences avoided when designing regions for deletion by allelic exchange. The allelic exchange experiments were performed using the vector pIMAY-Z (41). Deletions were repaired by allelic exchange using the wild-type allele. The primers used in these experiments are listed in Table S1. All mutants were subjected to whole-genome sequencing to assess whether additional mutations were unintentionally introduced during the allelic exchange procedure (Table S5).

**Normalizing bacterial suspensions for phenotype comparison.** One milliliter of overnight BHI broth culture was inoculated into 9 ml prewarmed BHI broth and incubated at 37°C with shaking at 200 rpm for an additional hour. The cultures were then washed twice with room temperature sterile saline (0.9% NaCl) and resuspended in the medium used for a specific assay. All bacterial suspensions were then normalized by dilution in media to an $OD_{600}$ of 0.147 ($\sim$1 × 10$^8$ CFU/ml).

**Growth curves.** Growth curve comparisons were performed for all isolates in this study. Ten microliters of normalized bacterial suspension grown in BHI broth were inoculated into 90 $\mu$l of tested media in a 96-well plate. Growth curves were performed in an EnSight plate reader (Perkin Elmer) with continuous shaking at 37°C. The $OD_{600}$ was measured every 10 min for at least 48 h. The media BHI, BHI with 1 mg/liter (0.5× MIC) and 1.5 mg/liter (0.75× MIC) linezolid were used as the test conditions. The growth curves were plotted using GraphPad Prism (v7.0d). The maximum doubling time was determined by fitting local regression over intervals of 1 h on growth curve data points and by taking the maximum value of the fitted derivative using the R package cellGrowth (www.bioconductor.org/packages/release/bioc/html/cellGrowth.html).

**Biofilm assay.** A static biofilm assay was performed on all isolates as described previously (42). Briefly, bacteria were cultured in plain TSB, TSB with 3% NaCl, or TSB with 1% glucose were used, with an inoculum of 5 × 10$^6$ CFU added to 200 $\mu$l of the media in 96-well plates, with incubation at 37°C for 18 h. The plates were sealed with adhesive plastic PCR film (MicroAmp Optical). After 18 h, the cultures were discarded, and the plates were washed four times with phosphate-buffered saline (PBS) using an enzyme-linked immunosorbent assay (ELISA) plate washer and dried at 65°C for 1 h. Residual bacterial biofilms adhering to the dried plates were then stained with 3% crystal violet for 5 min before washing four times with PBS. The plates were again dried at 65°C for 15 min. Two hundred microliters of 30% acetic acid was added to each well to resuspend the crystal violet, and the $OD_{590}$ was measured. Values were plotted and analyzed using GraphPad Prism (v7.0d).

**Whole-blood killing assay.** The whole-blood killing assay was adapted from a previously published method (43). Briefly, bacteria were washed twice with PBS and diluted to 4 × 10$^5$ CFU/ml, and then 125 $\mu$l of diluted bacterial suspension was added to 375 $\mu$l of freshly drawn human blood in heparinized tubes (1 × 10$^5$ CFU/ml). The mixture was incubated with agitation at 37°C for 4 h. Serial dilutions of the mixture in saline were plated onto sheep blood agar plates in triplicate. Bacterial survival ratios were calculated based on the CFU counts of the start point and endpoint of the experiments. The percentage bacterial survival was calculated by dividing endpoint colony counts by the starting point colony counts.

**Antibiogram profiling.** All isolates were assessed for antibiotic sensitivity using VITEK2 (bioMérieux). Vancomycin and linezolid macro E-tests were also performed on all isolates following standard methods (44). D-tests were conducted to verify inducible clindamycin resistance.

**Linezolid time-kill assay.** Linezolid time-kill assays were performed in BHI supplemented with 4 mg/liter linezolid. The normalized bacterial BHI suspensions were diluted to 1 × 10$^6$ CFU/ml. Ten microliters of bacterial suspensions (total 1 × 10$^4$ CFU) was added to 1 ml of 4-mg/liter linezolid BHI broth. The mixtures were sampled at 24 h after incubating at 37°C with shaking (200 rpm). The numbers of CFU of the mixture after treatment with linezolid were compared with the inoculum to determine the level of reduction.

**Growth competition assay.** Normalized bacterial suspensions of each sRNA knockout mutant and the wild-type parent strain JKD6009 were mixed at a ratio of 1:1. Five microliters of the mixture containing 1 × 10$^4$ CFU was inoculated into both 10 ml BHI broth and 10 ml BHI broth with 1 mg/liter linezolid, followed by incubation at 37°C with shaking (200 rpm) overnight. The genomic DNA from the mixed bacterial inocula and the resulting overnight cultures were extracted for quantitative assessment of mutant versus the wild type by digital droplet PCR assay (Bio-Rad). PCR primers were designed to generate amplicons of different sizes to differentiate the sRNA deletion strain and the parental strain, JKD6009. Digital droplet PCR was performed, and the data were analyzed as described by the manufacturer (Bio-Rad).

**Data availability.** Sequence reads for RNA-seq and verification of mutants are available at https://www.ncbi.nlm.nih.gov/bioproject/PRJNA576951.

## SUPPLEMENTAL MATERIAL

Supplemental material is available online only.

**FIG S1**, DOCX file, 0.3 MB.

**TABLE S1**, XLSX file, 0.01 MB.

**TABLE S2**, XLSX file, 0.1 MB.

**TABLE S3**, XLSX file, 0.01 MB.

**TABLE S4**, XLSX file, 0.01 MB.

**TABLE S5**, XLSX file, 0.01 MB.

## ACKNOWLEDGMENTS

This project was supported by the National Medical Research Council of Australia (GNT1105525 [T.P.S.] and GNT1105905 and GNT1026656 [B.P.H.]).

The funders had no role in study design, data collection and interpretation, or the decision to submit the work for publication.

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
