## [Reviewer comments · mSystems]

Comparative transcriptomic and functional assessment of linezolid-responsive sRNA genes in *Staphylococcus aureus*

Wei Gao, Romain Guérillot, Ya Hsun Lin, Jai Tree, Marie Beaume, Patrice Francois, Ian Monk, Torsten Seemann, Jacques Schrenzel, Benjamin Howden, and Timothy Stinear

Corresponding Author(s): Timothy Stinear, University of Melbourne

Review Timeline:

Submission Date:	October 13, 2019
Editorial Decision:	November 12, 2019
Revision Received:	November 18, 2019
Accepted:	November 23, 2019

Editor: Rup Lal

Reviewer(s): The following individuals involved in review of your submission have agreed to reveal their identity: Sarah Ben Maamar (Reviewer #1)

Transaction Report:

DOI: <https://doi.org/10.1128/mSystems.00665-19>

November 12, 2019

Prof. Timothy P Stinear
University of Melbourne
Dept. Microbiology and Immunology
Royal Parade
Parkville, Victoria 3010
Australia

Re: mSystems00665-19 (Comparative transcriptomic and functional assessment of linezolid-responsive sRNA genes in *Staphylococcus aureus*)

Dear Prof. Timothy P Stinear:

Below you will find the comments of the reviewers.

To submit your modified manuscript, log onto the eJP submission site at <https://msystems.msubmit.net/cgi-bin/main.plex>. If you cannot remember your password, click the "Can't remember your password?" link and follow the instructions on the screen. Go to Author Tasks and click the appropriate manuscript title to begin the resubmission process. The information that you entered when you first submitted the paper will be displayed. Please update the information as necessary. Provide (1) point-by-point responses to the issues raised by the reviewers as file type "Response to Reviewers," not in your cover letter, and (2) a PDF file that indicates the changes from the original submission (by highlighting or underlining the changes) as file type "Marked Up Manuscript - For Review Only."

Please return the manuscript within 60 days; if you cannot complete the modification within this time period, please contact me. If you do not wish to modify the manuscript and prefer to submit it to another journal, please notify me of your decision immediately so that the manuscript may be formally withdrawn from consideration by mSystems.

To avoid unnecessary delay in publication should your modified manuscript be accepted, it is important that all elements you upload meet the technical requirements for production. I strongly recommend that you check your digital images using the Rapid Inspector tool at <http://rapidinspector.cadmus.com/RapidInspector/zmw/>.

Sincerely,

Rup Lal

Editor, mSystems

Journals Department
Reviewer comments:

Reviewer #1 (Comments for the Author):

Excellent study and technically very sound and clear paper. Rigorous approach and well-structured manuscript. The study is definitely informative. I see no mandatory changes to require, the paper is clear and sounds.

However I have one question/remark for the authors:

I understand that the authors look at the effects related to the suppression of differentially regulated sRNA in bacteria exposed to linezolid on MRSA phenotypic antibio-resistance. Linezolid activity happens at the translational level. Linezolid actually occupies the A site of the ribosome and prevent tRNA from binding to the ribosome. In my understanding, sRNA are suspected to play a role in regulation of mRNA by binding with the mRNA and thus preventing the ribosome from binding to the mRNA. This regulation happens before or during the translation process similarly to the linezolid. I thus suspect that linezolid and sRNA activities are susceptible to interfere as they may happen both during mRNA translation. If this is the case, I would not expect any changes in phenotypic antibioresistance as linezolid activity would block the translation process and could mask sRNA regulation as mRNA wouldn't end up being translated in proteins anyway. Is my understanding correct? This logic is not necessarily obvious before learning about the study and its results, and this is what this study ended up suggesting to me. I believe the study brings interesting results. Have the authors noticed/investigated any difference in the mRNA and protein "levels" between bacterial mutants and wild type strains when they are not exposed to linezolid? Some interesting perspectives/future work could involve some proteomics and/or analysis of rRNA regulation as it could highlight interesting and complementary information on the distance effect of linezolid and mRNA down/up-regulations through sRNA.

If the authors have more information related to this question or if I misunderstood some aspects of this study, I definitely encourage the authors to add some statements/clarifications wherever it is suited to further comment or clarify the part I may have misunderstood. Nevertheless, I don't think this prevents from understanding the study.

Reviewer #2 (Comments for the Author):

This study describes the identification and experimental knockout of *S. aureus* sRNAs. This seems a quite reasonable way to generate hypotheses and subsequently test them. The performed experiments also generally seems technically competently in execution. I particularly commend the authors for doing the rescue control experiments and sequencing the knockout strains to get to the bottom of the decreased fitness.

Ultimately however, the KO strains seem to have no discernible phenotype difference. The authors provide some general rationale for why this might be, based on homeostasis, targeting overlap, small regulatory effects and the sensitivity and narrow scope of the phenotypic assays.

The study of sRNAs in general has been dominated by a few intensely studied examples, and many computationally predicted but not characterized sRNAs. I do believe there is some value in reporting these experiments despite their null effect. As the paper is, it is however very difficult to get any grasp on why there is no effect.

While there is nothing obviously unreasonable about the way the candidate sRNAs were selected it would be interesting to see plots ala 1E for the candidates and their putative targets. It's a little unclear to me if the figure only plots the reads mapped to the sRNA of interest, or it shows all reads in the neighbourhood, to attempt to ascertain if the expression was simply transcriptional noise, a byproduct of transcribing something nearby.

RESPONSE TO REVIEWER COMMENTS:

MS: mSystems00665-19

Title: Comparative transcriptomic and functional assessment of linezolid-responsive sRNA genes in *Staphylococcus aureus*

We thank the reviewers for their considered assessments. We offer our responses to questions raised below. The line numbers refer to the modified manuscript.

We trust you will find our responses satisfactory.

Sincerely,

Tim Stinear (on behalf of all the authors)

+++++

Reviewer #1

1. Linezolid actually occupies the A site of the ribosome and prevent tRNA from binding to the ribosome. In my understanding, sRNA are suspected to play a role in regulation of mRNA by binding with the mRNA and thus preventing the ribosome from binding to the mRNA. This regulation happens before or during the translation process similarly to the linezolid. I thus suspect that linezolid and sRNA activities are susceptible to interfere as they may happen both during mRNA translation. If this is the case, I would not expect any changes in phenotypic antibioresistance as linezolid activity would block the translation process and could mask sRNA regulation as mRNA wouldn't end up being translated in proteins anyway. Is my understanding correct?

Response: Yes, we agree with the reviewer's interpretation, but we planned our project around two hypotheses, based on our observation of the correlation between significant sRNA expression changes and linezolid exposure.

Hypothesis 1: Some sRNAs can promote linezolid resistance by regulation of specific mRNA transcripts and thus bypassing the linezolid resistance mechanism.

Hypothesis 2: Changes in expression of sRNA genes in response to linezolid treatment triggers a global transcriptional response that increases bacterial fitness under the protein synthesis blockade induced by the antibiotic.

Using our knockout mutants, hypothesis No.1 was disproved, since the linezolid MIC did not change in the mutants. However, hypothesis No. 2 covers many possible phenotypes that might counter the effect of linezolid, and experimental options are thus very wide. We used RNAseq and assays that tested bacterial fitness and some clinically relevant phenotypes. Our data indicated that the 18 sRNAs we have tested have no significant impact on bacterial fitness under linezolid exposure, in agreement with the premise

that linezolid is blocking translation process and so potentially masking sRNA regulation of translation. However, as the saying goes; an 'absence of evidence' is not 'evidence of absence', so additional experiments are clearly required here (see following response).

2. Have the authors noticed/investigated any difference in the mRNA and protein "levels" between bacterial mutants and wild type strains when they are not exposed to linezolid? Some interesting perspectives/future work could involve some proteomics and/or analysis of rRNA regulation as it could highlight interesting and complementary information on the distance effect of linezolid and mRNA down/up-regulations through sRNA

Response: We agree that these sRNAs may be involved in fine-tuning of responses to cellular stressors (eg antibiotics). As we conclude in our manuscript, to understand sRNA mechanism it will be critical that we examine sRNA-mRNA interactions (line 198).

Reviewer #2

1. While there is nothing obviously unreasonable about the way the candidate sRNAs were selected it would be interesting to see plots ala 1E for the candidates and their putative targets. It's a little unclear to me if the figure only plots the reads mapped to the sRNA of interest, or it shows all reads in the neighbourhood, to attempt to ascertain if the expression was simply transcriptional noise, a byproduct of transcribing something nearby.

Response: We have now included a supplementary figure (Figure S1), that includes the normalized read plots for all 18 sRNAs we investigated. These plots are showing all reads mapped to that region, not just those reads mapping to a particular sRNA gene.

November 23, 2019

Prof. Timothy P Stinear
University of Melbourne
Dept. Microbiology and Immunology
Royal Parade
Parkville, Victoria 3010
Australia

Re: mSystems00665-19R1 (Comparative transcriptomic and functional assessment of linezolid-responsive sRNA genes in *Staphylococcus aureus*)

Dear Prof. Timothy P Stinear:

Your manuscript has been accepted, and I am forwarding it to the ASM Journals Department for publication. For your reference, ASM Journals' address is given below. Before it can be scheduled for publication, your manuscript will be checked by the mSystems production editor, Ellie Ghatineh, to make sure that all elements meet the technical requirements for publication. She will contact you if anything needs to be revised before copyediting and production can begin. Otherwise, you will be notified when your proofs are ready to be viewed.

Sincerely,

Rup Lal
Editor, mSystems

Journals Department
Supplemental file 4: Accept
Supplemental file 6: Accept
Supplemental file 5: Accept
Supplemental file 3: Accept
Supplemental file 1: Accept
Supplemental file 2: Accept